# Homogeneous and Heterogeneous Catalytic Ozonation of Textile Wastewater: Application and Mechanism

**Magdalena Bilińska** [1,2,*], **Lucyna Bilińska** [1,2] **and Marta Gmurek** [1]

1    Department of Molecular Engineering, Lodz University of Technology, Wolczanska 213, 90-005 Lodz, Poland
2    Bilinski Factory of Colour, Mickiewicza 29, 95-050 Konstantynow Lodzki, Poland
*    Correspondence: magdalena.bilinska@dokt.p.lodz.pl; Tel.: +697826296

**Abstract:** This paper presents an overview of textile wastewater treatment by catalytic ozonation, highlighting the parameters of the process and accompanying mechanisms. Since more than 800,000 tons of dyes are produced annually and thousands of cubic meters of highly polluted textile wastewater have been emitted into the environment every day, this issue has become an environmental concern. Due to the high oxidative potential of ozone (2.08 V) and hydroxyl radical (2.80 V), the main reactive species in catalytic ozonation, the burdensome organic pollutants, including textile dyes, can be successfully decomposed. The paper shows the main groups of catalysts, emphasizing novel structural, nano-structured, and functionalized materials. The examples of catalytic ozonation in the industrial application for real textile wastewater were specially highlighted.

**Keywords:** textile wastewater treatment; advanced oxidation processes; catalytic ozonation; reaction mechanism; parameters of the ozonation catalytic process





## 1. Introduction

Thousands of cubic meters of highly polluted textile wastewater are emitted into the environment every day, making it a severe environmental threat. The average volume between 150 and 200 L per 1 kg of textiles produced in chemical processing is assumed [1]. However, according to Ghaly and co-workers, this value can be as high as 933 L per 1 kg in specific cases, e.g., in felt production [2]. Considering the ever-growing demand for textile goods, it is no wonder that textile wastewater has a significant impact on global pollution. China, India, Pakistan, Indochina countries, and some of EU countries are the major exporters of textiles [2]. Excluding Europe, where environmental protection is at a high technological level, most of these regions suffer from massive pollution. Australia's Perth region has an interesting approach to wastewater treatment for recycling purposes to meet long-term water needs. They use, among others, the method of oxidation ditches [3]. Membranes for closed water circulation at an on-site textile plant (Osmonics, USA) were also investigated to prevent clean water consumption [4]. There are several textile industrial zones in Bangladesh (such as Dhaka, Narayanganj, and Gazipur) reported being afflicted by untreated effluent discharge into water bodies [5]. The next region severely affected by the textile industry is the Southcoast area in China [6]. Heavy metals from dyes are reported to be the most burdensome pollution in local rivers [7]. The most shocking example of how the local landscape can be deformed by irresponsible wastewater management was shown in the Punjab region in India. Bhatia and co-workers reported the effects of direct textile wastewater discharge into surface water bodies, which made it a sewerage-like quality. What is more, the local community uses this water for living purposes, dramatically decreasing their hygienic standards [8]. Conclusively, numerous local legal acts uphold the limiting values of wastewater discharge [9]. At the same time, valid global institutions, such as the European Parliament [10], Organization for Economic Co-operation and Development (OECD) [11], or even public responsibility organizations such

as Greenpeace [12], took up efforts to decrease textile waste. This opens a wide field for investigation in the area of textile wastewater treatment. Especially when the priority is to make this wastewater a source of recyclable water for technological purposes as it stands in EU and OECD reports [10,11].

As far as textile wastewater is concerned, numerous treatments were used [13–15], and color removal was the most investigated. The color is the most vivid manifestation of pollution emitted by the textile industry into the water environment. Besides visual disgust, when emitted into natural water bodies, dyes' ability to light absorption disturbs living conditions by impairing water organisms' photosynthesis. Moreover, dyes' bioaccumulation and genotoxicity, exhibited in chromosomal aberrations, were proven for testing organisms [16]. Some azo dyes, such as Acid Black 077 [17], are confirmed as cancerogenic, and those were listed under the register of legally forbidden substances in EU countries [18]. However, environmental and health concerns are not the only pollution brought by textile wastewater. It has to be kept in mind that chemical textile processing on an industrial scale is a chain of many technological stages. Operations such as washing, scouring, bleaching, rinsing, dyeing, printing, fixing, and padding, following one by another, using a vast spectrum of chemicals. Therefore, numerous detergents, soda, acids, salts, urea, silicones, perfluorinated carbonates, enzymes, biologically active agents (e.g., silver, permethrin), and flame retardants can be used as textile auxiliaries in the production process, and finally occur in wastewater [9,19–21]. This wastewater is characterized by a high mineral content, resulting in low biodegradability (biodegradability index (BI) below 0.4) [22]. Consequently, bio-treatment is limited [13]. Textile wastewater is characterized by large ranges of chemical oxygen demand (COD): 150–12,000 (mg/L), biochemical oxygen demand (BOD): 80–6000 (mg/L) [23], and total organic carbon (TOC): 50–40,000 (mg/L) [13]. The textile wastewater purification is extremely difficult for additional reasons, as the chemical composition of the wastewater changes depending on the production settings. The settings of textile processing are imposed primarily by the composition of the textile garment. Each fiber type demands its dying technology and chemicals, which are presented in Table 1. This variety in the production system results in diversified wastewater characteristics changing over time.

**Table 1.** Examples of the most common textile fibers processing chemically in dye houses (own study of a different kind of textile treatment from Bilinski Factory sewage).

| Fiber Type | Type of Dye | Auxiliaries | Temperature (Celsius) of Wastewater | pH of Wastewater |
|---|---|---|---|---|
| Cellulosic (cotton, viscose) | Reactive Direct | dye<br>salt (NaCl, $Na_2SO_4$)<br>soda (NaOH, $Na_2CO_3$)<br>leveling agent<br>the enzyme (against $H_2O_2$) | 60, 80 | 10–12 |
| Woll, Silk, Polyamide | Acid | Dye<br>acid ($H_2SO_4$, formic, acetic)<br>salt ($Na_2SO_4$)<br>ammonium sulfate | 100 | 2–7 |
| Polyester | Disperse | Dye<br>Dispergator<br>acid (formic, acetic) | 130 (under pressure) | 5 |
| Acrylic | Cationic | Dye<br>acid (formic, acetic)<br>salt ($Na_2SO_4$, sodium acetate) | 90 | 4–5 |

This short characteristic of the textile processing system shows the complexity of the issue. The variety of pollutants which can occur besides dyes in textile wastewater, high temperature, low BI, and instability in time make it a challenge to find a proper treatment. This opens the discussion to novel enhanced strategies of textile-source pollutants removal

from water, making it recyclable. One of the last willingly presented approaches is combining chemical and biological treatments used as a sequence. This idea's motivation is to promote biodegradability and decrease toxicity at the first stage of chemical treatment to make further bio-treatment more efficient [13,14,24]. The preferable group of chemical treatments is advanced oxidation processes (AOPs). Ozone and reactive oxygen species (ROS) can decompose organics, including dyes, so ozone-based AOPs have been widely used in textile wastewater treatment [25–32]. Simultaneously, some traditional AOPs, $H_2O_2$/UV or Fenton, were insufficient for real textile wastewater purification because, despite discoloration, they were still characterized by high impurities [33–35].

Moreover, the application of hydrogen peroxide and UV-based technologies like $O_3$/$H_2O_2$, $O_3$/UV, and $O_3$/$H_2O_2$/UV for textile wastewater treatment is limited [33]. Since $H_2O_2$ is an expensive compound and the use of UV provides an additional energy cost of around EUR, USD $2/m^3$. The new trend in the treatment of textile wastewater is catalytic ozonation. The main objective of this review paper was to show perspectives on the use of catalytic ozonation for textile wastewater treatment. The presentation of vast catalysts is used for enhanced reactive oxygen species (ROS) production. The discussion on the possible mechanisms of catalytical action includes the main groups of catalysts. Particular emphasis was put on novel structural, nano-structured, and functionalized materials used as catalysts. Efforts to expose developments in reactor construction, ensuring proper contact between the catalyst, ozone, gas, and wastewater, were undertaken. The authors aimed to highlight examples of catalytic ozonation in the industrial application for real textile wastewater, showing its practical use.

## 2. Basic of Ozonation

Some facts from ozone chemistry must be brought up to understand the ozone-based catalytic processes. Ozone ($O_3$) is a strong oxidant, which can decompose numerous pollutants. Nevertheless, molecular ozone as an oxidant is selective, and some groups of pollutants, like carboxylic acids, naphthol, and phenol derivatives, were referred to as hardly degradable via single ozonation [36]. Moreover, it should be kept in mind that ozonation is usually carried out in the liquid phase, but ozone is applied in this process in its gaseous form. What is more, the $O_3$ molecule is unstable in the gas phase as well as in the liquid phase. As a consequence, two more issues arise: limited ozone absorption from the gaseous into the liquid phase and ozone self-decomposition [37,38].

The $O_3$ self-decomposition is one of the most relevant factors which define the treatment process. It was recognized as a chain of initiation, propagation, and termination steps. There are two of the most popular mechanisms which describe the $O_3$ self-decomposition. First, SBH (Staehelin, Buhler, and Hoigne), given by reactions (1)–(10) [39,40]:

Initiation:

$$O_3 + OH^- \rightarrow HO_2^\bullet + O_2^- \tag{1}$$

Propagation:

$$HO_2^\bullet \rightarrow H^+ + O_2^{\bullet-} \tag{2}$$

$$O_3 + O_2^{\bullet-} \rightarrow O_3^{\bullet-} + O_2 \tag{3}$$

$$O_3^{\bullet-} + H^+ \rightarrow HO_3^\bullet \tag{4}$$

$$HO_3^\bullet \rightarrow O_3^{\bullet-} + H^+ \tag{5}$$

$$HO_3^\bullet \rightarrow HO^\bullet + O_2 \tag{6}$$

$$HO^\bullet + O_3 \rightarrow HO_4^\bullet \tag{7}$$

$$HO_4^\bullet \rightarrow HO_2^\bullet + O_2 \tag{8}$$

Termination:

$$HO_4^\bullet + HO_4^\bullet \rightarrow H_2O_2 + 2O_3 \tag{9}$$

$$HO_4^\bullet + HO_3^\bullet \rightarrow H_2O_2 + O_3 + O_2 \tag{10}$$

The second most popular pathway of ozone decomposition is the TFG (Tomiyasu, Fukutomi, and Gordon) mechanism, in which the initial step starts by $OH^-$ ions, as in the SBH mechanism, or by $OH_2^-$ ions. This second mechanism was more relevant in an alkaline reaction medium [39,40].

Because the ozone molecule is non-polar, its solubility in water is limited. Therefore, $O_3$ concentration in the liquid phase is the next important factor during the ozonation process. When a bubble column is used for ozonation, the mass balance of ozone in the liquid phase can be described by Equation (11) [41]:

$$\frac{d\, C_L}{dt} = (k_L a)(C_L^* - C_L) - r_D \tag{11}$$

where $k_L a$ is the volumetric mass transfer coefficient in the liquid phase ($s^{-1}$), $C_L^*$ is the equilibrium molar concentration of ozone in the liquid phase (mol $m^{-3}$), $C_L$ is the molar concentration of ozone in the liquid phase (mol $m^{-3}$), and $r_D$ is the ozone decomposition rate (1000 mol $m^{-3}\ s^{-1}$).

All the above-mentioned ozone properties form a few consequences which should be taken into consideration when ozone-based treatment is planned. Firstly, according to the SBH model or TFG model, and other models as well, the decomposition of $O_3$ results in the formation of free radicals. This phenomenon is enhanced by alkaline pH. Therefore, two separate pathways of pollutants decomposition can be observed during ozonation: direct oxidation by the $O_3$ molecule and indirect reaction with the $HO^\bullet$ radical. In contrast to oxidation by the $HO^\bullet$, the reactions with $O_3$ are selective. $O_3$ can react with molecules at locations of high electron density by either cycloaddition, electrophilic substitution, or nucleophilic reactions [39]. Due to the presence of numerous double bonds in the structure of dye molecules, they can therefore likely be decomposed by $O_3$ molecules. However, low-molecular-weight by-products of dye decomposition are more likely decomposed by $HO^\bullet$ radicals, not $O_3$. Consequently, the accumulation of the by-products after basic ozone treatment is highly probable [42]. Secondly, the transferred ozone dose should be monitored during ozone-based treatment because this factor was found to be sensitive to any process modification, e.g., catalyst addition [43].

## 3. Catalytic Ozonation

Catalytic ozonation is widely concerned as an AOPs because of the production of hydroxyl radicals in the process [44]. The ozone self-decomposition resulting in hydroxyl radicals ($HO^\bullet$) production is referred to as a region of the catalytic action of catalysts most often [45]. However, this phenomenon is not a sufficient explanation of the crux of this process. Consequently, a deeper discussion on catalytic ozonation mechanisms should be raised, depending on the catalyst type. Generally, two systems of catalysis, homogeneous and heterogenous, are possible for catalytic ozonation, and both are discussed below (Figure 1). The first studies catalytic ozonation focused on metals and their various forms, such as salts, oxides, or deposits on the support [46]. The latest research focuses mainly on modern materials that combine the capabilities of catalysts from different groups. In a homogeneous system, ozone decomposition is initiated by transition metal ions, while in a heterogeneous system—by solid catalysts [44].

There are two mechanisms in homogeneous catalytic ozonation. The first leads to the formation of free radicals through the decomposition of ozone by metal ions [47]. The second is by the formation of complexes between the catalyst and the organic molecule, followed by oxidation of the resulting complex [48].

In heterogeneous catalytic ozonation, the catalytic activity of the catalyst is important, and for the catalyst to show it, there must be adsorption of ozone and an organic molecule on its surface. We have several catalytic ozonation decomposition mechanisms. The first is the decomposition of ozone on a reduced or oxidized form of metal deposited on the surface of the supporting material [49]. When metal oxides are used as catalysts, ozone

decomposition can occur on undissociated hydroxyl groups [50]. Ozone decomposition in the presence of activated carbon takes place at the primary centers of the catalyst [51].

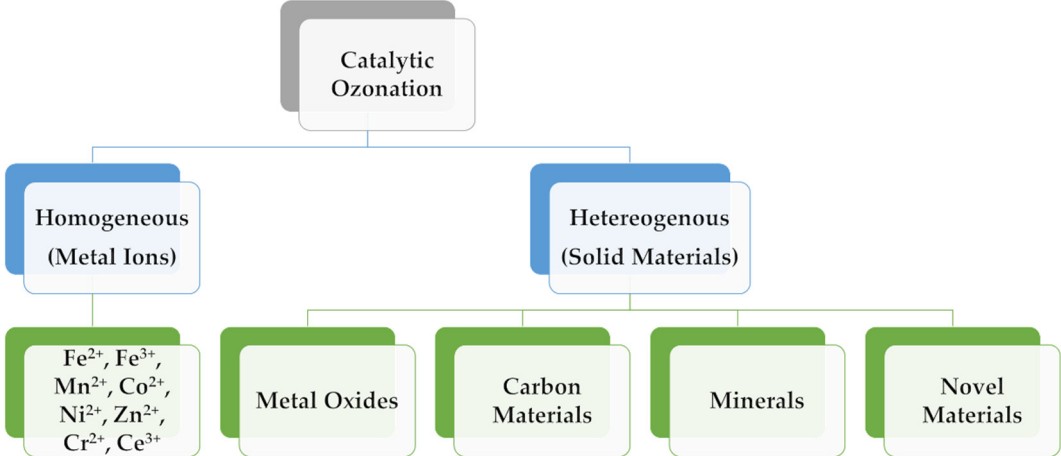

**Figure 1.** Classification of catalysts used in catalytic ozonation based on Nawrocki, 2010 [44].

The difference between the two methods is as follows [52]:

- Recovery of homogeneous catalysts is difficult and expensive, while in heterogeneous, it is easy and cheaper.
- Homogeneous shows poor thermal stability compared to heterogeneous.
- During homogeneous catalyst application, the selectivity is very good and focused on a single active site, while in heterogeneous, it is weaker, but more functioning for active centers.

Catalytic ozonation stands out from other purification methods in that it does not require additional costs (no UV lamps, often pH adjustment or removal of the resulting sludge) for the industry, and we often obtain a higher degree of mineralization [53]. In addition, the presence of the catalyst increases the number of hydroxyl radicals, i.e., it increases the speed of purification of the dye solution or textile wastewater than in the case of ozonation alone [54].

## 4. Homogeneous Catalytic Ozonation

The homogeneous system of catalytic ozonation can be catalyzed by transition metal cations, such as $Fe^{2+}$, $Fe^{3+}$, $Mn^{2+}$, $Zn^{2+}$, $Co^{2+}$ and $Ni^{2+}$ [55–58]. The possible catalytic activity of metal cations was referred to as indirect ozone decomposition of the complex of metal-pollutant [37].

The accompanying mechanisms of using various combinations of metallic ions with ozone to decolorize Reactive Red 2 have been studied by Wu et al. [55]. Table 2 shows the most effective parameters for the catalytic ozonation of a dye or wastewater. Low pH is most often used when using metal cations. In this group of catalysts, the most common system is the one based on the Fenton reaction $Fe(II)/O^3$ and $Fe(III)/O^3$ [55–57]:

$$Fe^{2+} + O_3 \rightarrow FeO^{2+} + O_2 \tag{12}$$

$$FeO^{2+} + H_2O \rightarrow Fe^{3+} + HO^\bullet + HO^- \tag{13}$$

$$Fe^{3+} + O_3 + H_2O \rightarrow FeO^{2+} + H^+ + HO^\bullet + O_2 \tag{14}$$

Using this method, it is important not to exceed the optimal dose of the catalyst, because $FeO^{2+}$ also oxidizes Fe(II) ions. The formation of radicals then occurs at low Fe(II) concentrations, reactions (15) and (16) start to work above the optimal dose and the reaction is inhibited [55]:

$$Fe^{2+} + HO^\bullet \rightarrow Fe^{3+} + HO^- \tag{15}$$

$$Fe^{2+} + FeO^{2+} + 2H^+ \rightarrow 2Fe^{3+} + H_2O \tag{16}$$

**Table 2.** Results of homogeneous catalytic ozonation.

| Wastewater Type/Pollutant | Catalyst Type | pH | $C_0$ (mg/L) | Conditions O₃ Dose (g/L) | Catal. Dose (g/L) | Color (%) | Removal COD (%) | TOC (%) | k (min⁻¹) | Year, Ref |
|---|---|---|---|---|---|---|---|---|---|---|
| Dye solution Reactive Red 2 | Fe(II), Fe(III), Mn(II), Zn(II), Co(II), Ni(II) | 2 | 100 | - | 0.0335 | - | - | 13–23 (after 5 min) | 1.299 1.278 3.296 1.015 0.843 0.822 | 2008, [55] |
| Textile Effluent | Fe(II), nZVI | - | - | 0.05–0.2 | 0.7 | 87 (after 40 min) | 73.5 (after 40 min) | - | 0.000751 0.000948 | 2018, [56] |
| Reactive Red 120 | Fe(III) | 3 | 100 | - | 0.00558 | - | 40 (after 30 min) | - | - | 2012, [57] |
| Reactive Red 2 | Mn(II) | 2 | 100 | - | 0.1 | 95 (after 5 min) | - | 17–21 (after 5 min) | - | 2008, [58] |

Studies by Malik et al. have shown that the use of $Fe^{2+}/O_3$ not only increases the process decolorization efficiency for dye and wastewater but will also increase the BI index from 0.26 to 0.4 for textile wastewater because catalytic ozonation accelerated the conversion of high molecular weight compounds into more biodegradable forms [56]. In their work, they also checked how zero valent iron nanoparticles (nZVI) work in the treatment of textile wastewater.

Another catalyst from the homogeneous group is $Mn^{2+}/O_3$, which also showed an efficient process of decolorizing the Reactive Red 2 dye solution [55,58]. Mn(II) reacts with ozone to form hydrated manganese oxide. The resulting oxide reacts with the dye, accelerating the discoloration process. Wu et al. have proposed the path of the reaction that may follow [58]:

$$Mn^{2+} + O_3 + 2H^+ \rightarrow Mn^{4+} + O_2 + H_2O \tag{17}$$

$$Mn^{4+} + 1.5O_3 + 3H^+ \rightarrow Mn^{7+} + 1.5O_2 + 1.5H_2O \tag{18}$$

$$Dye + Mn^{7+} + 1.5H_2O \rightarrow Mn^{4+} + 3H^+ + products \tag{19}$$

$$Mn^{2+} + Mn^{4+} \rightarrow 2Mn^{3+} \tag{20}$$

$$Mn^{2+} + O_3 + H^+ \rightarrow Mn^{3+} + O_2 + HO^\bullet \tag{21}$$

$$Mn^{3+} + O_3 + (Dye)^{2-} + H^+ \rightarrow Mn^{2+} + O_2 + HO^\bullet + products \tag{22}$$

$$Dye + HO^\bullet \rightarrow products \tag{23}$$

In the case of Zn(II), Co(II), and Ni(II), the lowest efficiency was achieved by nickel, regardless of the catalyst dose used [55]. The reaction mechanism for cobalt and nickel can be represented by the following reaction (24):

$$M^{(n-1)+} + O_3 + H^+ \rightarrow M^{n+} + HO^\bullet + O_2 \tag{24}$$

The use of homogeneous catalysts for the decolorization of dye solutions or treatment of textile wastewater is not very frequent and it may be due to the lack of a developed method of removing ions from the solution after the process. Often homogeneous catalysts are used for preliminary research into later heterogeneous catalysts [48,58–60].

## 5. Heterogeneous Catalytic Ozonation

Heterogeneous systems in catalytic ozonation are solids, so they can be removed from the solution in a simpler way than homogeneous systems. There are three probable mechanisms of operation of heterogeneous catalytic ozonation, as shown in Figure 2 [44]. In this system, the adsorption capacity of the catalyst is critical as at least one of the reagents must be adsorbed onto the catalyst surface. Correspondingly, the process's catalytic action is possible when one of the presented activities occurs: ozone is adsorbed on the catalyst

surface. A pollutant molecule is adsorbed on the catalyst, and both ozone and pollutants are adsorbed on the surface.

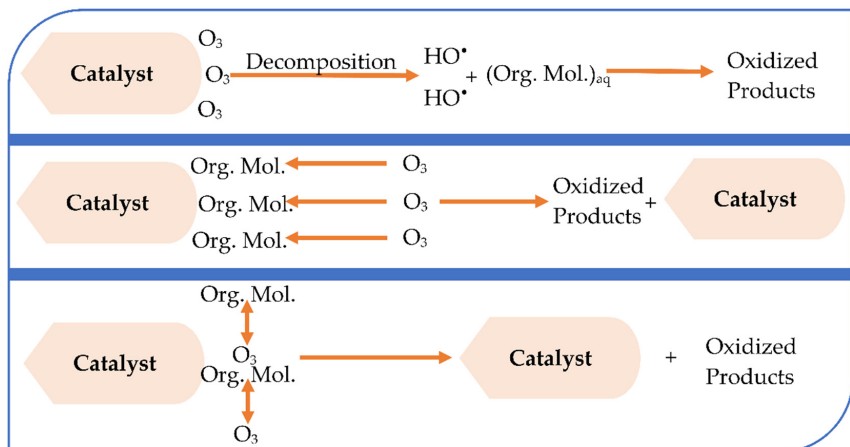

**Figure 2.** Different mechanisms of heterogeneous catalytic ozonation are based on Nawrocki, 2010 [44].

The main groups of substances used as heterogeneous catalysts are metal oxides, carbon species, minerals, and new materials. The further part of the article will present the use of particular groups for the discoloration of the dye solution as well as textile wastewater, along with the test parameters for which the best results were obtained, along with the test parameters for which the best results were obtained.

*5.1. Metal Oxides*

For the treatment of textile wastewater or dye solutions, metal oxides such as $Al_2O_3$ [61,62], $CeO_2$ [63], $MgO$ [64–66], $MnO_2$ [58,67], $CoO$ [67–69], $FeO_3$ [70,71], $Ca(OH)_2$ [72] were used. When using metal oxides, it must be remembered that the pH value has a great influence on the properties of their surface, and hence on their efficiency in removing contaminants. Surface properties that can change are the isoelectric point, the specific surface area, and the Lewis acid sites (accept electrons) [44].

Polat et al. showed that the effectiveness of catalytic ozonation with the use of alumina depends on the surface of the catalyst and the pH of the solution [61,62]. The dye molecules are absorbed on the $Al_2O_3$ surface, therefore the larger the surface area, the greater the adsorption capacity [62]. They are degraded by radicals on the catalyst surface and by ozone in the solution. Conducting the process at acidic pH resulted in a low concentration of hydroxide ions and significantly reduced the share of ozone decomposition into $HO^{\bullet}$ [61].

$CeO_2$ was another catalyst used for dye discoloration [63]. Efficient dye removal and COD reduction was increased significantly with the addition of the catalyst to the process, a phenomenon that can be explained by the fact that $CeO_2$ promotes the decomposition of ozone to form reactive oxygen species (such as $HO^{\bullet}$). Qiu et al. analyzed the mechanism of organic matter removal using $CeO_2$. First, the dye and ozone are adsorbed on the surface of the catalyst, then the catalyst initiates the decomposition of ozone into hydroxyl radicals. The resulting radicals attack organic substances, creating $H_2O$ and $CO_2$ [63].

MgO has also been used to decolorize dye solutions [64–66]. Moussavi et al. obtained a maximum COD and color removal at pH 8, hydroxyl radicals are mainly generated at pH above 10, and the obtained results are explained by the presence of MgO which could have contributed to lowering the optimal pH or led to the generation of other radicals [65]. Increasing the pH did not lead to an increase in the efficiency of the process. The results obtained at an alkaline pH can be ascribed to an increase in the availability of hydroxide ions and an accelerated breakdown of ozone. This translates into the formation of hydroxyl radicals and MgO-hydroxyl radicals. The mechanism is adopted that the catalyst is also the initiator of radical formation, similar to $CeO_2$, and it adsorbs ozone on its surface. Similar

results were obtained by Chokshi et al. where the best results were obtained for pH 7 [64]. The use of MgO also has advantages for environmental reasons, it is environmentally friendly, recyclable, and easy to separate [66].

Wu et al. used $MnO_2$ in their research on the discoloration of the dye solution. They presented the mechanism of operation in this process (25)–(27). Ozone reacts with the hydroxide ions on the catalyst surface according to the following reactions to form hydroxyl radicals on the catalyst surface [58,67]:

$$MnO_2 + H_2O \rightarrow MnO_2 - OH^- + H^+ \tag{25}$$

$$MnO_2 - OH^- + O_3 \rightarrow MnO_2 - OH^\bullet + O_3^- \tag{26}$$

$$MnO_2 - OH^\bullet + dye \rightarrow MnO_2 + products \tag{27}$$

Cobalt oxide occurs mainly in the form of $Co_3O_4$. According to Faria et al., the increased mineralization of pollutants achieved by catalytic ozonation is mainly due to the presence of $HO^\bullet$ formed as a result of catalytic ozone decomposition [67]. On the other hand, Chokshi et al. put forward the theory that cobalt oxide works according to the third mechanism, where the catalyst absorbs not only ozone but also pollutions [69].

Fe-shaving-based catalyst was used for the catalytic ozonation of industrial wastewater in the research conducted by Li et al. In both studies, an analysis of the existing mechanisms was carried out, where the formation of hydroxyl radicals was confirmed. In this process, the absorption of water and ozone takes place on the catalyst surface, where water is dissociated and ozone is decomposed [70,71].

The last representative of metal oxides used in the purification of the dye solution is $Ca(OH)_2$. Quan et al. proposed a mechanism to improve the catalytic ozonation process. During the mineralization of organic pollutants, $HCO^{3-}$ and $CO_3{}^{2-}$ ions are formed, which act as radical scavengers that negatively affect the efficiency of the process. $Ca(OH)_2$ dissociates into $Ca^{2+}$ and $HO^-$ in an aqueous solution. The resulting cation reacts with $HCO^{3-}$ and $CO_3{}^{2-}$ ions to form $CaCO_3$. The hydroxide anion reacts with ozone in the $O_3/HO^-$ system. These two processes taking place simultaneously lead to an increase in the degradation efficiency and the mineralization of organic pollutants [72].

Table 3 summarizes the best process parameters using metal oxides for catalytic ozonation.

The use of metal oxide catalysts, due to their easy removal from the solution, is more common than their homogeneous counterparts. Comparing the test results from both groups, it can be noticed that by using metal oxides, better mineralization of organic pollutants can be obtained, which is associated with more complex mechanisms accompanying the purification process.

## 5.2. Carbon Materials

Another group of heterogeneous catalysts used in catalytic ozonation, distinguished by a large specific surface area (from 500 $m^2/g$ to 1500 $m^2/g$), adsorption capacity, chemical resistance, no metal leaching [51,73,74], relatively low price, and the possibility of recycling [66,75,76] is activated carbon.

In studies on catalytic ozonation with the use of activated carbon, it has been proven that the presence of this catalyst in solution initiates the radical decomposition of ozone in water, leading to the formation of hydroxyl radicals. Research has shown that metal centers, basal plane electrons, and functional groups, such as chromene, pyrene, and pyrrole, are active centers in the production of $HO^\bullet$ radicals from ozone decomposition [75,77,78]. Mostly, the catalytic decomposition of ozone with the use of activated carbon takes place in the alkaline pH range (6–10) [51,74,77,79,80].

Wang et al. discovered that granular activated carbon (GAC) used for catalytic ozonation of textile wastewater can be successfully regenerated and reused for 20 days of a continuous process. The obtained results indicated more than twice greater efficiency after the regeneration process [76].

**Table 3.** Examples of catalytic ozonation with metal oxides.

| Wastewater Type/Pollutant | Catalyst Type | pH | C$_0$ (mg/L) | O$_3$ Dose (g/L) | Catal. Dose (g/L) | Color (%) | COD (%) | TOC (%) | k (min$^{-1}$) | Year, Ref |
|---|---|---|---|---|---|---|---|---|---|---|
| Textile Wastewaters (Basic Blue 41, Basic Yellow 28, Basic Red 18.1) | Al$_2$O$_3$ | 4 | - | 0.0918 | 5 | - | Blue Dye—45.3; Red/Yellow Dyes—100 | - | - | 2015, [61] |
| Acid Red 151; | Al$_2$O$_3$ | 2.5 | 200 | 0.0147 | 5 | 98.4 (after 30 min) | 78.7 (after 30 min) | - | 0.136 | 2008, [62] |
| Remazol Blue R | | | | | | 98.3 (after 30 min) | 82.6 (after 30 min) | - | 0.132 | |
| Lemon Yellow | CeO$_2$ | 6 | - | - | 0.5 | - | - | 97 | - | 2014, [63] |
| Reactive Black 5 | MgO | 7 | - | - | 0.05–2 | 99.9 | - | 38.8 | - | 2016, [64] |
| Reactive Red 198 | MgO | 8 | - | - | 1–6 | 100 | 69 | - | - | 2009, [65] |
| Methylene Blue | MgO | 9 | 500 | - | - | 77 (after 60 min) | 12.15 (after 60 min) | - | 0.025 | 2016, [66] |
| Reactive Red 2 | MnO$_2$ | 2 | 100 | - | 0.8 | 95 (after 5 min) | - | 17–21 (after 5 min) | - | 2008, [58] |
| CI Acid Blue 113, aniline | Mn-O, Co-O, Ce-O | 3, 5.5, 6 | - | 0.05 | 0.35 | - | - | ~80 | - | 2009, [67] |
| Naphtol Blue Black | Co$_3$O$_4$, | 5 | - | - | 0.1 | | | 10–30 | - | 2007, [68] |
| Reactive Black 5 | Co$_3$O$_4$ | 7 | - | - | 0.2 | 99.99 | 80 | - | - | 2017, [69] |
| Industrial Wastewater | Fe-shaving based | 6.81 ± 0.34 | - | 0.065 | 3 | - | - | 98.5 | - | 2018, [70] 2019, [71] |
| Acid Red 18 | Ca(OH)$_2$ | | - | 0.065 | 3 | ~100 | | ~100 | - | 2017, [72] |

To increase the efficiency of the process, attempts were made to increase the surface of activated carbon and change the morphology of the catalyst surface [80]. The more developed surface area (BET surface area and pore size) contributed to an increase in the absorption of the dye on the catalyst surface and an increase in discoloration.

Mahmoodi et al. investigated the performance of multi-walled carbon nanotubes (MWCNTs) in catalytic ozonation. This material, thanks to its unique mechanical, thermal, chemical, and electrical properties can be an interesting alternative to activated carbon. The mechanism of catalytic ozonation should be similar to that of activated carbon due to the similarity in bonding and structure [81]. Qu et al. also used carbon nanotubes for catalytic ozonation of the Indigo. The nanotubes have been enriched with -COOH functional groups. The presence of the catalyst contributed to the increase in mineralization. In these studies, it was concluded that the mechanism of catalytic ozonation may be different, however. The special nanostructure of carbon nanotubes can lead to the storage of ozone in a microstate and lead to an increase in direct electron transfer, translating into an increase in the efficiency of the process [82].

For catalytic ozonation, Faria et al. used a composite of activated carbon and cerium oxide. In this way, they obtained a catalyst that can be used three times without any change in the efficiency of the process. It has been noted that the mechanism of catalytic ozonation using an activated carbon-cerium oxide composite includes both surface reactions that occur when using only activated carbon and that are aqueous solution reactions involving HO$^\bullet$ that occurred during catalytic ozonation with cerium oxide [83].

The next carbon-based catalyst is carbon aerogel, which has a very extensive surface and high porosity [84,85]. The mechanism proposed by Hu et al. contains an increase in ozone mass transfer to the water phase due to the porosity of the catalyst. It also promotes the breakdown of ozone into reactive forms [84]. According to Niu et al., the addition of MnO$_2$ to the graphene aerogel leads to an increase in the efficiency of the decomposition of organic pollutants by increasing the density of the electron cloud of Mn atoms deposited on the aerogel, it favors the decomposition of ozone and the generation of HO$^\bullet$ radicals [85].

Copper(II)-doped carbon dots were used for catalytic ozonation of dyes as well as real textile wastewater [86]. The use of polyethylene glycol (PEG) in the catalyst synthesis made it possible to adsorb the dye on the surface of the dots. The addition of copper increased the active surface of the carbon dots and the ozone adsorbed on the surface is catalytically transformed into hydroxyl radicals, which then react with the adsorbed dye.

Table 4 summarizes the best process parameters using carbon materials as catalysts.

**Table 4.** Examples of catalytic ozonation with the application of carbon materials.

| Wastewater Type/Pollutant | Catalyst Type | pH | C$_0$ (mg/L) | Conditions O$_3$ Dose (g/L) | Catal. Dose (g/L) | Color (%) | Removal COD (%) | TOC (%) | k (min$^{-1}$) | Year, Ref |
|---|---|---|---|---|---|---|---|---|---|---|
| Reactive Black 5 | Activated carbon | 11.29 | - | 0.9 | 0.005 | - | 40 (after 60 min) | 35 (after 60 min) | - | 2020, [75] |
| Aniline | Activated Carbon | 3 7 9 | 13.2 | 50 | 0.35 | - | - | - | 0.188, 0.290, 0.233 | 2007, [79] |
| Real textile effluent | Activated carbon | 8.5 | 100 | 0.00754 | 0.3 | ~100 (15 min) | - | 20.7 (after 60 min) | 0.47 | 2005, [87] |
| Reactive Blue 194 | Granular activated carbon | 7 | 200 | 0.179 | - | 100 (after 20 min) | - | - | | 2020, [51] |
| Basic Blue 9 | Granular activated carbon | 10 | - | - | 2 | 80 (after 5 min) | 68 (after 5 min) | - | 0.72 | 2013, [77] |
| Reactive Red 195 | Granular activated carbon | 11 | 100 | - | 1 | 90.4 (after 2 min) | - | - | - | 2012, [80] |
| Acid Red 3R | Granular Activated Carbon | 7 | 100 | 0.0417 | 2 | - | - | 78.2 (after 60 min) | - | 2018, [74] |
| C.I. Reactive Red 194, C.I. Reactive Yellow 145 | Granular Activated Carbon | 6.3 5.9 | 100 | 0.035 | 10 | ~100 (after 30 min) | 80 (after 30 min) | 50 (after 30 min) | - | 2009, [78] |
| Methylene Blue | Granular Activated Carbon | 9 | 500 | - | - | 63 (after 60 min) | 25.36 (after 60 min) | - | 0.016 | 2016, [66] |
| Bio-treated dyeing finishing wastewater | Regenerated granular activated carbon | - | - | 0.0185 | 2 | 81.7 (after 25 min) | 71 (after 25 min) | - | - | 2019, [76] |
| Reactive Red 198, Direct Green 6 | Multiwalled carbon nanotube | 3 | 150 | - | 0.03 | 100 (after 16 min), 100 (after 20 min) | - | - | - | 2013, [81] |
| Indigo | Carbon nanotubes functionalized with carboxyl groups | 4 | 100 | 0.141 | 0.005 | ~99 (after 20 min) | - | 35.1 (after 2 h) | −0.219 | 2015, [82] |
| CI Acid Blue 113, CI Reactive Yellow 3 CI Reactive Blue 5 | Activated carbon–cerium oxide | 5.6 | 50 | 50 | 0.35 | 97 88 98 (after 5 min) | - | 88 (after 120 min) | - | 2009, [83] |
| Dyeing effluents | Carbon aerogel-Co$_3$O$_4$ | 7, 10 | - | 0.008 | 3 | 100 (after 10 min) | 80 (after 30 min) | - | - | 2016, [84] |

**Table 4.** *Cont.*

| Wastewater Type/Pollutant | Catalyst Type | pH | $C_0$ (mg/L) | $O_3$ Dose (g/L) | Catal. Dose (g/L) | Color (%) | COD (%) | TOC (%) | k (min$^{-1}$) | Year, Ref |
|---|---|---|---|---|---|---|---|---|---|---|
| | | | | **Conditions** | | | **Removal** | | | |
| Rhodamine B | Graphene/$\alpha$-MnO$_2$ nanocrystals hybrid aerogel (GMA) | - | 50 | 0.035 | - | 100 (after 60 min) | 89.02 (after 15 min) | - | 0.2859 | 2019, [85] |
| Methyl Orange Real textile effluent | Copper(II)-doped carbon dots | 7 | - | 1.98 | - | 99.8 (after 6 min) 41 (after 60 min) | - | - | 1.184 0.012 | 2022, [86] |

Combining ozonation with carbonaceous materials is effective in removing color and some organic matter from highly colored wastewater. Despite the popularity of, for example, carbon nanotubes and their wide application, most research focuses on the use of granulated activated carbon, which is easily available and achieves good performance in purifying dye solutions.

### 5.3. Minerals

The use of natural minerals as catalysts for catalytic ozonation is still not widely used in the treatment of textile wastewater. The use of natural-based materials may limit the application of synthetic catalysts, which leads to decreased production often harmful to the environment and associated with high costs [88]. Natural minerals can be used as catalyst supports [89,90] or catalysts [91,92]. Due to their functionality and abundance of resources, they are relatively cheap equivalents and may in the future become substitutes for currently used catalysts in environmental protection [92]. Moreover, due to their high porosity and specific surface area, they can show greater adsorption and thus higher efficiency in the catalytic ozonation process [89].

Montmorillonite (Mt) is an aluminosilicate that Boudissa et al. modified to give acid-base and hydrophobic properties. In the first modification, Na was deposited on the Mt surface, and in this variant, the dye molecules were absorbed by hydrophobic interactions. In the second modification, Fe(II) was used, and cation exchanges and mobility of $Fe^{2+}$ to catalytic activity took place here. The last modification was the activation of the surface with acid, and the surface showed the lowest hydrophilic character, increasing the mineralization of the dye [89].

Valdes et al. used volcanic sand [91] and natural zeolite [93] for catalytic ozonation of the dye solution. Metal oxides in volcanic sand generate hydroxyl groups on their surface, which are involved in the decomposition of ozone. Volcanic sand accelerates the self-decomposition of ozone, which means it accelerates the generation of active forms [91]. The mechanism of catalytic ozonation with zeolite is the same as with volcanic sand because hydroxyl groups are also formed on its surface [93].

Taseidifar et al. first treated natural magnetite ($Fe_3O_4$) with oxygen plasma (chemical etching) and argon plasma (sputtering effect), resulting in greater surface roughness. Thanks to this modification, more active sites, and better mass exchange were obtained. On the magnetite surface, hydroxyl groups are formed, which later take part in ozone decomposition, as in the case of earlier minerals [94].

For the catalytic ozonation of the dye solution, Moussavi et al. used raw (27 $m^2$/g) and calcine (34 $m^2$/g) magnetite. Calcination led to the destruction of pyrite and the transformation of magnetite ($Fe_3O_4$) into hematite ($Fe_2O_3$). The mechanism of the catalytic ozonation reaction with the participation of magnetite is similar to the previous minerals. The greater efficiency of the calcined catalyst can be attributed to $Fe_2O_3$, which is its main component and has a higher catalytic potential than $Fe_3O_4$ [88].

Dong et al. used natural brucite for catalytic ozonation of a dye solution, which consists mainly of $Mg(OH)_2$ and dopants of Si, Ca, and Fe oxides. An increase in pH occurs during the reaction due to the dissociation of magnesium oxide, therefore the possibility that the reaction mechanism is homogeneous here due to $OH^-$, although there is solid brucite powder in the solution, was highlighted [92].

The last presented mineral used for catalytic ozonation of the dye solution is the aluminosilicate Montanit300®, and although it is not very active by itself, it gains greater activity after modification with $H_2SO_4$ [90]. Inchaurrondo et al. found that there are active sites (Mn) on the surface of the mineral, acid treatment increased the Si: Al ratio, the specific surface of the catalyst increased, and the percentage of iron and manganese decreased. The improved catalytic properties of the modified zeolite can be explained by an increase in the Si: Al ratio and an increase in the hydrophobicity, which in such minerals is responsible for the reactions between ozone adsorption and organic pollutants on the catalyst surface.

Table 5 summarizes the best process parameters using minerals for catalytic ozonation.

**Table 5.** Examples of catalytic ozonation with minerals.

| Wastewater Type/Pollutant | Catalyst Type | pH | $C_0$ (mg/L) | Conditions O$_3$ Dose (g/L) | Catal. Dose (g/L) | Removal Color (%) | COD (%) | TOC (%) | k (min$^{-1}$) | Year, Ref |
|---|---|---|---|---|---|---|---|---|---|---|
| Methylene Blue, Methyl Green, Methyl Orange, Methylthymol Blue | Ion-exchanged montmorillonite, (NaMt and Fe(II)Mt), crude bentonite, and acid-activated counterparts (HMt) | - | 200 | 0.0083 | 0.04 | - | - | - | - | 2019, [89] |
| Methylene Blue | Volcanic sand | 8 | 30 | 0.006 | 50 | 70 (after 50 min) | - | - | 0.09 | 2010, [91] |
| Methylene Blue | Zeolite Volcanic sand | 2 | - | - | 15 | - | - | - | 0.054 0.12 | 2009, [93] |
| Basic Blue 3 | Natural Magnetite modified with argon plasma | 6.7 | 90 | 0.003 | 0.6 | 93.47 (after 15 min) | - | - | 0.1814 | 2015, [94] |
| Reactive Red-120 | Raw and Calcined magnetite | 11 | 100 | - | 3 | - | - | 96.1 (after 120 min) | 0.082 | 2012, [88] |
| Active Brilliant Red X-3B | Brucite | - | 500 | - | 0.5 | 89 (after 15 min) | 32.5 (after 15 min) | - | - | 2007, [92] |
| Orange II | Aluminosilicate, Montanit300 | 6 | 100 | - | 1 | 100 (after 240 min) | - | 91 (after 240 min) | - | 2021, [90] |

Cheap solid materials such as zeolites, aluminosilicates, sand, brucite, and magnetite have been used for the heterogeneous ozonation of dyes. The use of natural catalysts is a great advantage from an ecological point of view because production processes that can deliver toxic compounds to the environment are avoided and costs are reduced. However, the use of mineral catalysts for the treatment of textile wastewater is still a little-explored area, which in time may turn out to be one of the main paths that scientists will follow. The disadvantage of using natural minerals is their wearability during the catalytic ozonation process, as the oxides on their surfaces often dissociate [88,92].

*5.4. Novel Materials*

Modern catalytic materials strive to increase the effectivity of catalytic ozonation by changing the properties of the catalyst surface, such as porosity, pore-volume, or mechanical strength. When creating such catalysts, these already proven materials are often combined to obtain a product that exhibits even better properties: composites [95,96], metals, or metal oxides on support [62,97–102]. Increasing the mesoporosity of the material or its delamination provides more active sites where catalytic reactions can take place [66,103,104]. Then, such surfaces can be modified, thanks to which we obtain a larger catalytic surface [105–110].

Chokshi et al. used the composites La-Co-O (La$_2$O$_3$/Co$_3$O$_4$) and Ag-La-Co to discolor the Reactive Black 5 solution [95,96]. The obtained Ag-La-Co composite is characterized by the simplicity of preparation by the co-precipitation method, which is quick and simple and does not require the use of any organic solvent. The mechanism of catalytic ozonation with the use of these composites can be described in several simultaneous reactions (Figure 3). First, ozone passes from the gas to the liquid phase, and there is adsorption of ozone and organic substances on the catalyst surface and chemical reactions between them, reactions between the adsorbed substances and HO$^\bullet$ radicals, and reactions of by-products [96].

Alumina was used as a carrier for perfluorooctylic acid (PFO), and the resulting catalyst was used to decolorize the Acid Red 151 (AR-151) and Remazol Blue R (RBBR) solution in the catalytic ozonation process [62]. Eriol et al. proved that the percentage by weight of PFO on the support did not have a large impact on the degree of discoloration, but it contributed to the increase in COD reduction concerning the alumina alone. These results were explained by the presence of a significant amount of PFO acid on the surface which produced longer alkyl chains that increased perfluorooctyl alumina (PFOA) activity. The adsorption capacity of PFOA also improved over alumina alone due to the increased

hydrophobicity of the catalyst surface. The formation of perfluorinated alkyl chains on the surface promotes the change of character from acidic/basic to hydrophobic. This phenomenon is attributed to the easier adsorption of organic molecules to the catalyst surface. The obtained tests showed that the efficiency of this catalyst largely depends on the pH of the solution and the surface properties as well as the nature of the dye. Therefore, for 100% PFOA, the best results were obtained with RBBR at pH 13.

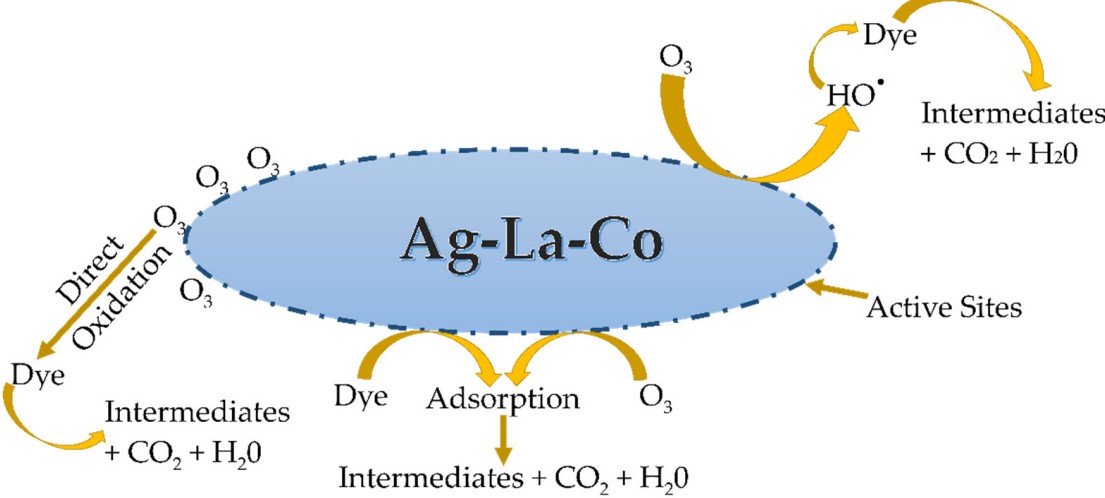

**Figure 3.** Proposed mechanisms of action of Ag-La-Co catalyst based on Chokshi, 2021 [96].

Asgari et al. produced carbon-doped magnesium oxide that was additionally doped with the powdered, acid-free eggshell membrane (C-MgO-EMP). The obtained catalyst was used to decolorize textile wastewater. The doping of activated carbon will allow an increase in the degradation of pollutants in catalytic ozonation by increasing the number of radicals. From the obtained results, it was concluded that the performance of the catalyst improved the degree of mineralization of the byproducts, and the improvement of the results was attributed to better interactions and reduced resistance to mass transfer. Due to the extensive surface of the catalyst, organic pollutants and ozone are absorbed on the surface undergoing mineralization [97].

Ghuge et al. investigated the process of ozonation of the catalytic dye solution Reactive Orange 4 and textile wastewater, using two catalysts Cu/SBA-15 and Ru-Cu/SBA-15. SBA-15 is a mesoporous material with a highly ordered structure, characterized by uniform pore size and a parallel arrangement. Its catalytic activity was improved by immobilizing metallic copper on its surface, creating new active sites [98]. Using the t-butanol radical scavenger, it was found that the decomposition of ozone corresponds to the removal of color from the solution, while the demineralization takes place using hydroxyl radicals which are formed by the reaction of ozone with hydroxyl groups formed on the catalyst surface [99]. Another example of bimetallic nanostructures used to treat dye wastewater is Fe/Mn@$\gamma-Al_2O_3$ produced by Liang et al. Through studies of the mechanism, they found that the use of a catalyst increases the constant reaction rate of removing dye wastewater compared to ozone alone, due to the increased efficiency of HO$^\bullet$ and $^1O_2$ formation [102].

The chip catalyst, i.e., porous copper fibers (PCFSS) sintered together and then loaded with Cu/Zn/Al/Zr metal oxide catalysts, was used to decolorize Basic Yellow 87 in the catalytic ozonation process [100]. The obtained catalyst can be used many times without losing its catalytic activity. After 10 uses, 99.2% dye discoloration and 58.9% COD removal were achieved. PCFSS has a three-dimensional lattice structure with high porosity and a specific surface area, facilitating mass transfer and promoting multiphase reactions. Zhu et al. noticed a slight loss of the catalytic chip mass, the reason for this may be the dissolution of metal oxides on the support surface in water. This phenomenon may contribute to receiving H$^+$ ions from the solution. The mechanism of chip operation was

proposed by adsorbing ozone and organic matter on its surface, and then the interaction of ozone with metal oxides. This leads to obtaining both the radicals and the reactants on the surface of the catalyst, facilitating the reaction between them [100]. A similar adsorption/catalytic effect of a ceramic membrane with deposited Mg, Ce, and Mn oxides was used by Li et al. The membrane uses two techniques: filtration and catalytic ozonation. Coating the membrane with metal oxides increased the efficiency of pollutant removal from 30% to 80% [101].

The Mg-O-SC$_{CA-Zn}$ granular catalyst was made of synthesized porous sludge-derived char and magnesium hydroxide and was used for catalytic ozonation of an aqueous solution of Methylene Blue [66]. Kong et al. attributed the catalytic ability of the resulting catalyst to the combination of MgO nanoparticles deposited on the surface and SC$_{CA-Zn}$ porosity.

Lu et al. prepared a magnetic mesoporous catalyst for the catalytic ozonation of Acid Orange II [103]. Spinel ferrites are soft magnetic materials with a stable structure and the general formula $MeFe_2O_4$. Due to their magnetic property, they can be easily removed after the process. The decomposition of ozone into active radicals is controlled by electron transfer between ozone on the catalyst surface and $MgFe_2O_4$. The obtained results indicate the presence of the $HO^\bullet$ radical mechanism and the adsorption of both dye and ozone on the catalyst surface along with the occurrence of surface reactions [103].

El Hassani et al. looked at Ni-based double hydroxide (Ni-LDH) layered nanomaterials and used them for the catalytic ozonation of Methyl Orange. The mechanism with the use of the Ni-LDH catalyst consists in adsorbing ozone on its surface and creating weak bonds with the hydroxyl groups present in each layer of the catalyst. Nickel, on the other hand, acts as an active site for ozone decomposition. Additionally, dye absorption takes place on the catalyst surface [104].

Another example of a catalyst used to decolorize the Methyl Orange dye solution is nanohybrid $NiFe_2O_4$-NiO growing on porous nickel 3D (NF) foam [105]. In heterogeneous catalytic ozonation, it has shown high stability and activity, and most importantly the catalyst is recycled. Numerous active sites are created on the catalyst surface, which enables the formation of $HO^\bullet$ and $O_2^{\bullet-}$ radicals, accelerating the decomposition of $O_3$. Due to the porous structure of the catalyst, there is a rapid liquid flow and mass transfer during catalytic ozonation with its use. The used support material is flexible, which allows it to be shaped for the needs of future reactors and easier to remove from them.

Hien et al. used ground zinc slag (Zn-S) in the process of catalytic ozonation of the Direct Black 22 (DB22) dye solution. Zinc slag consists of calcium and zinc which increase the degradation of $O_3$. The degradation of the dye occurs through the mechanism of reaction with hydroxyl radicals and adsorption on the catalyst surface. The structure of the catalyst is characterized by large pores and an extensive surface, thanks to which its use has resulted in better mineralization and discoloration of the dye. DB22 is an anionic dye with sulfo groups that can form a complex with the $Ca^{2+}$ ($CaCO_3$) ions present. This further increases the catalyst's ability to absorb the dye from the solution [106].

An example of improving the already existing catalyst parameters is the work of Pereira et al. where calcium alginate beads incorporated with nZVI were made [107]. Using nZVI is associated with self-aggregation and passivation of nanoparticles, reducing their reactivity and reaction area. Immobilizing them in an alginate shell overcomes these limitations. However, the use of this catalyst involves the use of pneumatic agitation, due to their poor shear resistance, and the use of conventional mixers could lead to breakage. Figure 4 shows the proposed catalyst mechanism in the catalytic ozonation of Reactive Red 195. Sone et al. focused on the green synthesis of ZnO nanoparticles, using an aqueous extract of the palm tree Doum. Their research enables more ecological catalyst production with equally good yields [111].

Pervez et al. developed hydrophilic electrospun membranes based on polyvinyl alcohol (PVA) using low-temperature synthesis. The obtained catalyst was used for the catalytic ozonation of Methylene Blue (MB). PVA membranes are characterized by chemical and thermal stability and have numerous hydroxyl groups on their surface. The produced

membrane has a hydrophilic surface, which results in strong electrostatic interactions between it and the cationic Methylene Blue (greater adsorption on the catalyst surface) [108].

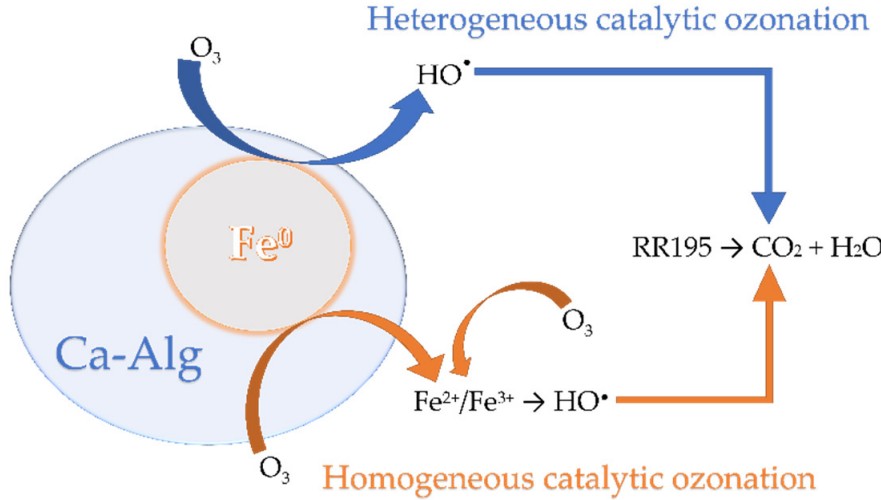

**Figure 4.** Proposed mechanism of catalytic ozonation of Reactive Red 195 dye using nZVI based on Pereira, 2021 [107].

Chen et al. prepared bimetallic sulfur-doped yttrium copper and yttrium oxide by the co-precipitation method [109]. The obtained S-CuYO catalyst was used for the catalytic ozonation of aniline solution. Due to the synergy between the forms $\equiv$Cu(I) and $\equiv$Y(III), the obtained catalyst is characterized by a large number of hydroxyl groups on its surface, a large specific surface area, and a large pore size. The dominant reactive oxygen species are HO$^\bullet$ radicals, in addition, electrons are transferred from S-CuYO to dissolved oxygen, which creates $O_2^{\bullet-}$. The research also discovered that during the process of catalytic ozonation, $H_2O_2$ is produced, which will turn into HO$^\bullet$ [109].

Metal-organic frameworks are characterized by very good adsorption properties. Obtained by Yu et al., the MIL-53(Fe) catalyst showed high catalytic activity due to its porous surface (mass-transfer properties) and active surface area of $372.65 \text{ m}^2/\text{g}$. The study of the mechanism accompanying the use of the MIL-53(Fe)/$O_3$ system proved that the active site is conducive to ozone decomposition. The main ROS here are $HO_{ads}^\bullet$, $O_2^{\bullet-}$, and $O_2^1$. Additionally, the reaction rate using this catalyst increases almost 13 times [112].

Faghihinezhad et al. created an O@g-$C_3N_4$/$Al_2O_3$ nanocatalyst for catalytic ozonation of textile wastewater. The catalyst consists of magnetic oxidized g-$C_3O_4$ nanoparticles and $Al_2O_3$ immobilized on their surface. Within 60 min of the process, color reductions of 99% and COD reductions of 77% were obtained. Moreover, the regenerative capabilities of the obtained catalyst were checked and after 5 cycles, the COD efficiency decreased by only about 8% [110]. The mechanism of catalytic ozonation here is as follows: oxidation by $O_3$ on both surfaces, oxidation by free radicals HO$^\bullet$ generated from the decomposition of $O_3$.

Javed et al. used broken laboratory-grade borosilicate glass and coated it with cobalt (Co-BSG). The resulting catalyst decolorized the Methylene Blue solution in 8 min to 92% and removed the COD in 40 min to 93% [113]. Thanks to the use of recycled glass, this method is not only more environmentally friendly but also economically beneficial. The reaction mechanism using this catalyst takes place through the interaction of $O_3$ with the surface of the catalyst. The radical mechanism is dominant here.

The Table 6 summarizes the best process parameters with the use of a given catalyst.

**Table 6.** Examples of catalytic ozonation with novel materials.

| Wastewater Type/Pollutant | Catalyst Type | Conditions | | | | Removal | | | k (min$^{-1}$) | Year, Ref |
|---|---|---|---|---|---|---|---|---|---|---|
| | | pH | C$_0$ (mg/L) | O$_3$ Dose (g/L) | Catal. Dose (g/L) | Color (%) | COD (%) | TOC (%) | | |
| Reactive Black 5 | La-Co-O | 3 | 100 | - | 0.005 | 99 (after 30 min) | - | - | - | 2020, [95] |
| Reactive Black 5 | Ag-La-Co | 7 | 100 | - | 0.5 | 95 (after 10 min) | - | 60 (after 80 min) | 0.0727 | 2021, [96] |
| Acid Red 151; Remazol Blue R | 100% perfluorooctyl alumina | 13 | 200 | 0.0147 | 5 | 98.8 (after 30 min), 97.4 (after 30 min) | 75.7 (after 30 min), 96.6 (after 30 min) | - - | 0.132, 0.158 | 2008, [62] |
| Textile wastewater | C-doped MgO eggshell C-MgO-EMP | - | - | 0.08 | 0.23 | 93 (after 10 min) | - | 78 (after 10 min) | 1.545 | 2019, [97] |
| Reactive Orange 4 | Cu/SBA-15 | 9 | 100 | 0.005 | 2 | 100 (after 21 min) | - | 86 (after 60 min) | 0.031 | 2018, [98] |
| Reactive Orange 4 | Mesoporous | 9 | | | | 100 (after 21 min) | 70.4 (after 60 min), | | | 2018, [99] |
| Dye Industrial Effluent | Ru-Cu/SBA-15 | | 100 | 0.005 | 2 | 100 (after 21 min) | 90 (after 4 h) | - | - | |
| Dye wastewater | Fe/Mn@γ−Al$_2$O$_3$ | 7.5 7 | - | 0.006 | 0.2 | - | - | - | 0.132 | 2022, [102] |
| Basic Yellow 87 | Porous copper fiber sintered sheet | 6.6 | 216 | 0.5 | - | 100 (after 120 min) | 60 (after 120 min) | - | - | 2014, [100] |
| Azo Dye 4BS | Mg-Ce membrane, Mg-Mn membrane | 8.5 | 12 | 0.012 | - | 85 (after 5 min), 88 (after 5 min) | - | >75 (after 5 min), >75 (after 5 min) | - | 2021, [101] |
| Methylene Blue | G-MgO-SC$_{Ca-Zn}$ | 9 | 500 | - | - | 98 (after 60 min) | 50 (after 60 min) | - | 0.04 | 2016, [66] |
| Acid Orange II | MgFe$_2$O$_4$ | 5 | 50 | 0.005 | 0.1 | - | - | 48.1 (after 40 min) | - | 2016, [103] |
| Methyl Orange (MO) | Ni-LDHs | 9 | 500 | - | 1 | 96 (after 60 min) | 72 (after 60 min) | - | 0.053 | 2019, [104] |
| Methyl Orange | NiFe$_2$O$_4$-NiO/NF | 6.84 | - | 0.0041 | - | 96 (after 60 min) | - | 71 (after 60 min) | - | 2021, [105] |
| Direct Black 22 | Zn-S | 3–11 | 100 | 3.38 | 0.75 | 99 (after 20 min) | 82 (after 25 min) | - | - | 2020, [106] |
| Reactive Red 195 | nZVI-Alg | 3.0–6.5 | 25 | 0.008 | 50 | 100 (after 30 min) | 98 (after 90 min) | - | - | 2021, [107] |
| Coralene Rubine-S2G | ZnO-400 nanoparticles | 6.8–8.4 | 130 | - | 0.050 | 100 (after 35 min) | - | - | 0.359 | 2021, [111] |
| Methylene Blue | Polyvinylalcohol (PVA) nanofibrous membranes | 3.02 | 20 | - | 0.03 | 94 (after 60 min) | - | - | - | 2020, [108] |
| Aniline | S-CuYO | 3–11 | 10 | - | 0–2.0 | 96 (after 15 min) | - | 57.7 (after 15 min) | - | 2020, [109] |
| Rhodamine B | Fe-MOFs | 7 | 40 | - | 0.2 | 100 (after 2.5 min) | - | 40 (after 30 min) | 5.76 | 2019, [112] |
| Real textile wastewater | O@g-C$_3$N$_4$/ Al$_2$O$_3$ | 7.1 | - | - | 0.5 | 99 (after 60 min) | 77 (after 60 min) | - | 0.155 | 2022, [110] |
| Methylene Blue | Co-BSG | 6.8 | 30 | - | 5 | 92 (after 8 min) | 93 (after 40 min) | - | - | 2022, [113] |

Modern materials can over time replace the most commonly used catalysts, such as TiO$_2$ or activated carbon. They are gaining popularity because by modifying their structure [100,111], surfaces [104,110,112], or composition [62,66], it is possible to improve their catalytic abilities, obtaining better and better pollutant removal abilities. The modifications also overcome the limitations of already existing catalysts [107].

## 6. Conclusions

Despite many studies, it is still unclear how ozone adsorbs on the catalyst surface. Many articles claim that the decomposition of ozone takes place on the catalyst surface, leading to the formation of free radicals. It is also questionable whether the adsorption of organic substances on the catalyst surface plays a key role in the catalytic process itself. It is certain, however, that the parameters of the process have a great influence on the catalytic efficiency of the degradation of limited pollutants. Therefore, it is important to study the mechanisms of catalytic ozonation to better understand the phenomena that occur during this process and the main factors influencing its efficiency.

The important advantages of using catalytic ozonation to purify colored solutions and textile wastewater are:

- Increasing the degradation of pollutants in water, mainly organic.
- Supporting the mineralization of organic compounds.
- Reduces ozone consumption compared to the ozonation process itself. When reviewing the parameters used in the studies with the best results, some relationships can be noticed:
- The pH of the solution affects the charge of active centers located on the catalyst surface and the ionic charge of organic molecules. This parameter is responsible for the interaction between the catalyst and the impurities. Low pH slows down the ozone decomposition process, which contributes to the longer contact time of ozone with pollutants, but from the industrial point of view, it is less profitable.
- An important parameter due to the increased interest in environmental protection and costs for the company is the stability of the catalyst and the possibility of its reuse.
- By increasing the amount of the catalyst, we provide more active sites, contributing to the decomposition of ozone, i.e., increasing the reactive radicals in the solution.
- Increasing the ozone flow rate also increases the generation of reactive radicals. This parameter is limited by the number of active sites on the catalyst surface.
- Increasing the ozone dose increases gas permeation into the dye solution/sewage, thus improving its availability to react with pollutants. Increasing the ozone dose is also associated with higher production costs.

Despite many studies conducted on the use of catalytic ozonation to purify dye solutions, there is still a lack of research focused on textile wastewater. Textile wastewater is more complex and contains additional substances which makes it more difficult to clean. Additionally, it is important to pay attention to the financial aspect so that the developed method of textile wastewater treatment is beneficial for the industry.

**Author Contributions:** Conceptualization, M.B. and L.B.; methodology, M.B., L.B. and M.G.; validation, M.B., L.B. and M.G.; investigation, M.B., L.B. and M.G.; data curation, M.B.; writing—original draft preparation, M.B.; writing—review and editing, L.B. and M.G.; supervision, M.G.; project administration, L.B. All authors have read and agreed to the published version of the manuscript.

**Funding:** The research was funded by the Norway Grants 2014–2021 via the National Centre for Research and Development under grant number NOR/SGS/TEX-WATER-REC/0026/2020-00.

**Acknowledgments:** Special thanks to Textile Company Biliński, Konstantynów Łódzki, Poland for their cooperation.

**Conflicts of Interest:** The authors declare no conflict of interest.

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
