# Peer review of "Homogeneous and Heterogeneous Catalytic Ozonation of Textile Wastewater: Application and Mechanism"

_catalysts, doi:10.3390/catal13010006_

Round 1

Reviewer 1 Report

The review has been appropriately compiled. Its English is acceptable, although it needs some improvement from the viewpoints of both composition and grammar (e.g., usage of commas).

Some abbreviations should be resolved in the case of the first occurrence, e.g., BI index, nZVI.

The presentation concept is suitable, applying tables for comparison and texts for the description of the characteristic details, regarding the same works. The review may discuss a higher number of more recent works.

Some points to be improved or corrected are indicated in the manuscript by yellow highlights and remarks.

Reviewer 2 Report

Catalytic ozonation for textile wastewater treatment – review

Overall comments:

This is a review article on fundamental research on catalytic ozonation for textile wastewater treatment. The article is divided into 6 main parts including:

1.     Introduction

2.     Basic of ozonation

3.     Catalytic ozonation

4.     Homogeneous catalytic ozonation

5.     Heterogeneous catalytic ozonation

6.     Conclusions

The paper shows the main groups of catalysts, emphasizing novel structural, nano-structured, and functionalized materials. The examples of catalytic ozonation in the industrial application for real textile wastewater were specially highlighted.

The method of AOPs using catalytic ozonation is well known these days, and is known as catazone for short. The catazone decomposition process is very suitable for textile dyeing wastewater because of their high pollutant contents and very difficult to decompose. Personally, I feel this is a nice and interesting review.

Specific comments:

1.     The title of the article is large, while the article only focuses on reviewing and evaluating different catalysts including homogenous and heterogeneous. On the other hand, the paper also only focuses on the issues related to the mechanism of catalytic ozonation. Therefore, the title of the article should be more specific.

2.     Introduction section: presented about the pollution level of textile wastewater, it is necessary to introduce the content values of pollution parameters such as COD, BOD, TOC. In the introduction, there is a general description of textile wastewater in the world, but it only focuses on some countries such as China, India, Pakistan, Indochina countries and some EU countries, which are the major exporters of textiles. Excluding Europe, where environmental protection is on a high technological level, most of these regions suffer from massive pollution. However, it is also necessary to review more about textile production and wastewater treatment measures of some other developed countries in the US and Australia. On the other hand, it is also necessary to describe about the advantages of the catalytic ozonation method compared to other methods.

3.     The article should cite important references such as:

-        B. LegubeN. K. Leitner, 1999, Catalytic ozonation: a promising advanced oxidation technology for water treatment, Catalysis Today, 53, 61-72. This looks a historical paper, as the first people discovered catalytic ozonation.

-        Rame Rame, Purwanto Purwanto and Sudarno Sudarno, 2020, Potential of Catalytic Ozonation in Treatment of Industrial Textile Wastewater in Indonesia, Jurnal RisetTeknologi Pencegahan Pencemaran Industri, 11, 1-11. This reference review is very close to this article.

-        Jianlong Wang 1, Hai Chen, 2020, Catalytic ozonation for water and wastewater treatment: Recent advances and perspective, Sci Total Environ, 704:135249.

4.     Line 40: “such as European”, insert “the” before “European”

5.     Line 54: “However, days arise environmental and health concerns; are not the only…”, should remove “days arise” and “;”

6.     Line 103, “burned” should be changed “kept”

7.     Lines 117 and 180, please correct into “homogeneous”

8.     The words “vulcanic” in line 383 and table 5 page 13 should be changed into “volcanic”

9.     Line 533, insert “the” before “palm tree Doum”

Conclusions:

Overall, this review is pretty well written. But the focus is only on the catalytic ozonation reaction mechanism, so the title of the paper should be specific.
